# Incidence of sudden cardiac death in the young: a systematic review

Keith Couper [1,2] Oliver Putt,[1] Richard Field,[2] Kurtis Poole,[1] William Bradlow,[3] Aileen Clarke,[1] Gavin D Perkins,[1,2] Pamela Royle,[1] Joyce Yeung [1,2] Sian Taylor-Phillips [1]

¹Warwick Medical School, University of Warwick Warwick Medical School, Coventry, UK
²Critical Care Unit, University Hospitals Birmingham NHS Foundation Trust, Birmingham, UK
³Department of Cardiology, University Hospitals Birmingham NHS Foundation Trust, Birmingham, UK

**Correspondence to**
Dr Keith Couper;
k.couper@warwick.ac.uk

## ABSTRACT

**Objective** To summarise studies describing incidence of sudden cardiac death in a general population of young individuals to inform screening policy.

**Design** Systematic review.

**Data sources** Database searches of MEDLINE, EMBASE and the Cochrane library (all inception to current) on 29 April 2019 (updated 16 November 2019), and forward/backward citation tracking of eligible studies.

**Study eligibility criteria** All studies that reported incidence of sudden cardiac death in young individuals (12–39 years) in a general population, with no restriction on language or date. Planned subgroups were incidence by age, sex, race and athletic status (including military personnel).

**Data extraction** Two reviewers independently assessed study eligibility, extracted study data and assessed risk of bias using the Joanna Briggs Institute critical appraisal checklist for prevalence studies.

**Analysis** Reported incidence of sudden cardiac death in the young per 100 000 person-years.

**Results** 38 studies that reported incidence across five continents. We identified substantial heterogeneity in population, sudden cardiac death definition, and case ascertainment methods, precluding meta-analysis. Median reported follow-up years was 6.97 million (IQR 2.34 million–23.70 million) and number of sudden cardiac death cases was 64 (IQR 40–251). In the general population, the median of reported incidence was 1.7 sudden cardiac death per 100 000 person-years (IQR 1.3–2.6, range 0.75–11.9). Most studies (n=14, 54%) reported an incidence between one and two cases per 100 000 person-years. Incidence was higher in males and older individuals.

**Conclusions** This systematic review identified variability in the reported incidence of sudden cardiac death in the young across studies. Most studies reported an incidence between one and two cases per 100 000 person-years.

**PROSPERO registration number** CRD42019120563.

## INTRODUCTION

Cardiovascular disease is the leading cause of death worldwide.[1] In the young, however, deaths due to cardiovascular disease are much less frequent than deaths due to other causes such as unintentional injury, suicide and homicide.[2] Nevertheless, over 20 000 young individuals aged under 45 die due

### Strengths and limitations of this study

► This systematic review is reported in accordance with the Preferred Reporting Items for Systematic Reviews and Meta-Analyses checklist.
► We identified studies that reported the incidence of sudden cardiac death in the young through searches of key databases and citation tracking.
► All eligible studies were included, irrespective of the publication date or publication language.
► We identified and described variability in the definition of sudden cardiac death and methods used to identify sudden cardiac death across studies.
► We did not include cases of out-of-hospital cardiac arrest with successful resuscitation.

to cardiovascular disease in the USA each year.[3] The subgroup of young cardiovascular deaths that occur suddenly has a particularly profound effect on the individual's family and local community.

Screening of asymptomatic individuals for cardiac conditions, such as cardiomyopathies and channelopathies, has been proposed as a strategy to reduce the incidence of sudden cardiac death in young people. The best evidence that such a strategy might be effective comes from a single Italian before-after study that reported a decreased incidence of sudden cardiac death in athletes following the introduction of a mandatory athlete screening programme.[4] In view of the limitations of the current evidence, screening of young asymptomatic non-athletes is not presently supported by either the American Heart Association or European Society of Cardiology (AHA/ESC).[5–7]

For organisations that make decisions regarding the implementation of population screening programmes, such as the US Preventive Services Task Force and UK National Screening Committee, a clear understanding of the incidence of the target condition provides important context for decision-making.[8 9] In particular, incidence

is an indicator of the size of the health problem, and maximum benefit which could be gained from screening, to be balanced against the potential harms such as stopping young people from participating in the exercise. Previous reviews of sudden cardiac death incidence have focused on the incidence across all ranges, or in specific populations.[10 11] The aim of this systematic review is to describe current evidence on the incidence of sudden cardiac death in the young.

## METHODS

We conducted a systematic review of studies that report the incidence of sudden cardiac death in the young. We were originally commissioned by the UK National Screening Committee to undertake a rapid review of incidence studies.[12] We subsequently chose to develop the rapid review into this systematic review, with no limitation on study location, date or publication language.

The review is written in accordance with the Preferred Reporting Items for Systematic Reviews and Meta-Analyses guidelines.[13]

### Search and study selection

We searched MEDLINE (1946–current), EMBASE (1947–current) and the Cochrane library (inception–current) for eligible studies. The search strategy was developed by an information scientist. We used a combination of keywords and MESH terms to describe the population (eg, young adult, adolescent), condition (eg, sudden death, sudden cardiac death) and study type (eg, cohort, longitudinal) of interest. An example search strategy is included in the electronic supplement.

Following searches and duplicate removal, two reviewers independently screened study titles and abstracts. Conflicts were resolved through discussion or, where needed, arbitration by a third reviewer. The same process was adopted for review of full-text papers. We identified additional studies through forward and backward citation tracking of included studies.

### Study eligibility criteria

We included studies that described the incidence (per 100 000 person-years) of sudden cardiac death in the young, or that provided sufficient data to allow calculation of the incidence. We defined a young individual as someone aged 12–39 years. This age range reflects the target group for a sudden cardiac death programme that was recently considered by the UK National Screening Committee.[12] The lower age cut-off of 12 years was used in the Italian sudden cardiac death screening study.[4] For reasons of pragmatism, we included studies where the reported incidence combined individuals in our target age range with younger individuals (≥1 year). Exclusion of individuals under 1 year avoided the risk of conflating sudden cardiac death with sudden infant death syndrome. We excluded studies where the reported incidence combined individuals in our target age range with

older individuals (≥40 years), due to the marked increase in sudden cardiac death incidence from the age of 40.[14]

We also excluded studies that reported incidence only in a population with previously known disease, or only included deaths that occurred at certain times of day (eg, at work or school) or during specific activities (eg, sports). We imposed no restriction on language or publication date. We did not mandate a specific definition of sudden cardiac death or case-ascertainment process, but these were recorded during data extraction and considered in the risk of bias assessments.

Our main population of interest was the general population aged 12–39. To be included in the main analysis, studies were required to report incidence in a general population that included males and females with an age group that spanned at least ten-years between 12 and 39 years. Where studies reported more than one age subgroup (eg, 12–21 years; 22–31 years), we selected the subgroup closest to the age of mid-late teens to be included in the main analysis. Our planned subgroups included incidence by age, sex, race and athletic status (including military personnel). We do not report subgroups of subgroups (eg, athletes broken down by sex).

### Data extraction and analysis

We used a piloted pro forma to extract population characteristics, study design case identification methods, use of systematic screening and outcome data. We assessed risk of bias using the Joanna Briggs Institute critical appraisal checklist for prevalence studies.[15] We developed review-specific criteria for each checklist question. Two reviewers independently extracted data and assessed risk of bias, with discrepancies resolved through discussion. Papers published in a language other than English were translated by a fluent speaker.

We extracted incidence per 100 000 person-years and associated 95% CI from the papers. Where these data were not reported, we calculated them from the reported number of sudden cardiac deaths and person follow-up years. We computed the 95% CI based on a binomial distribution, using Stata V.15.1. Clinical heterogeneity in relation to population characteristics and case ascertainment methods precluded pooling of data.

### Patient/ and public involvement

We did not involve patients or members of the public in setting the research question, designing the study, undertaking the study, the interpretation of the results or study write-up.

## RESULTS

We performed initial database searches on 29 April 2019, and updated searches on 16 November 2019. Through searches and citation tracking, we identified 8360 unique citations of which we reviewed the full-text of 238. We excluded 27 of the 65 papers initially included due to

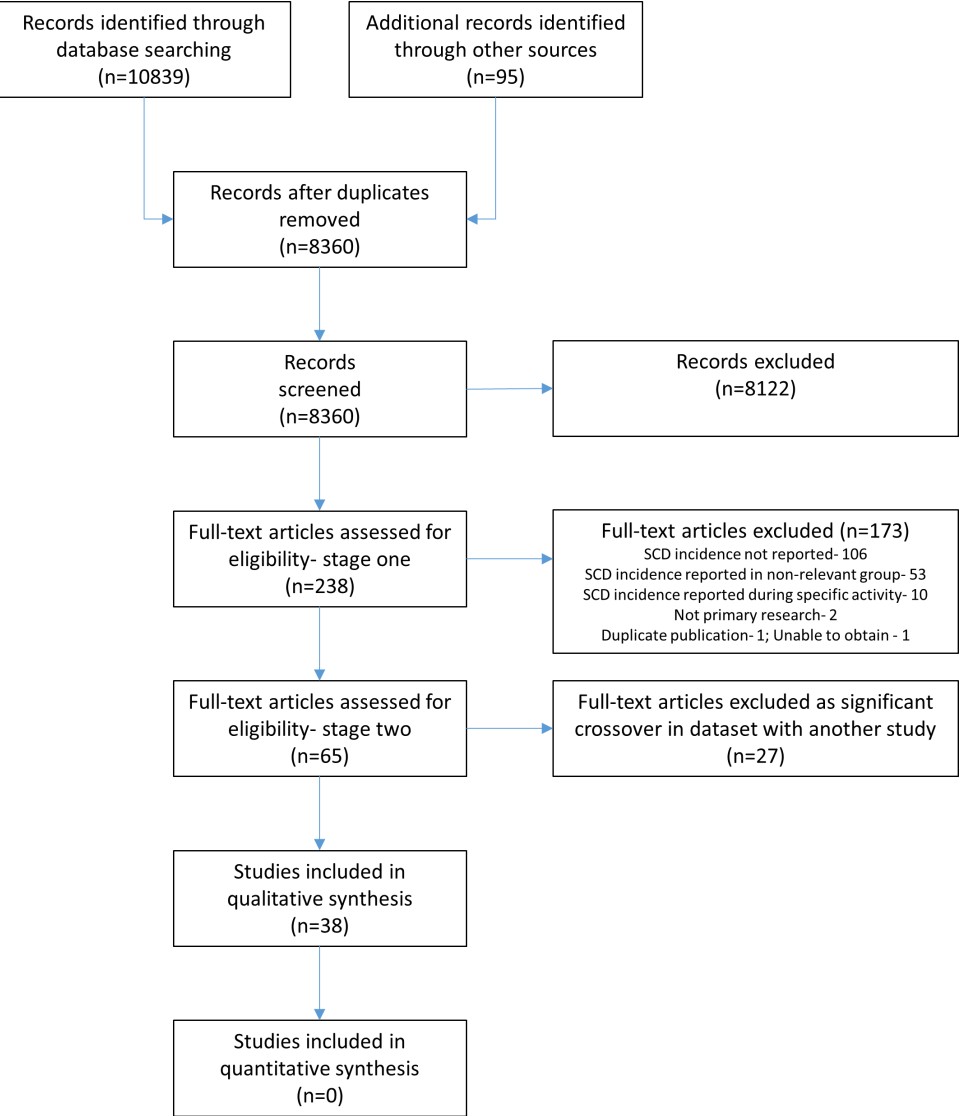

**Figure 1** Preferred Reporting Items for Systematic Reviews and Meta-Analyses flow chart of study selections. SCD, sudden cardiac death.

substantial or complete data overlap with other studies (figure 1).

Most of the 38 included studies[4 16–52] were undertaken in Europe (n=20, 53%) and North America (n=14, 37%) (table 1). The majority were retrospective studies (n=29, 76%). Median number of deaths and person follow-up years were 64 (IQR 40–251) and 6.97 million (IQR 2.34 million–23.70 million), respectively. Study duration ranged from 1 year to 41 years.

There was variability in the population, case ascertainment methods, and sudden cardiac death definition used between studies. This methodological heterogeneity precluded meta-analysis. Only nine studies (24%) stated that they used the AHA/ESC definition of sudden cardiac death. Our risk of bias assessment found that most studies were at high risk of bias. (figure 2; online supplemental table S1) As shown in figure 2, the main factor contributing to a high risk of bias was the method used to identify cases of sudden cardiac death. In five

(13%) studies, sudden cardiac death was identified using death certificates. The remaining studies used a range of approaches that typically incorporated autopsy data to some extent.

Our main analysis included 26 studies (figure 3). The median of reported incidence was 1.7 sudden cardiac death per 100 000 person-years (IQR 1.3–2.6, range 0.75–11.9). Of the 26 studies, 17 (65%) reported an incidence of less than two sudden cardiac death events per 100 000 person-years, and 14 (54%) studies reported an incidence between one and two cases per 100 000 person-years. All four studies reporting an incidence over three cases per 100 000 person-years were at high risk of bias across multiple domains.

Geographically, incidence appeared highest in African and Asian countries, although this may in part reflect the case ascertainment process and associated risk of bias. These studies also tended to include only older individuals (≥18 years).

**Table 1** Summary of included studies

| | Design | Setting | Population: type/age | SCD cases | Follow-up years | Systematic screening | SCD definition | Case ascertainment | |
| --- | --- | --- | --- | --- | --- | --- | --- | --- | --- |
| | | | | | | | | Identification of deaths | Main method to determine SCD |
| Anastasakis et al[16] | PC | Greece (2002–2010) | General (1–35) | 226 | 12750000 | No | AHA/ESC | Government records/death certificates | Autopsy record review |
| Anderson et al[51] | RC | USA (1977–1988) | General (5–39) | 183 | 9836196 | No | Other | Medical examiner records | Autopsy record review |
| Asatryan et al[45] | RC | Switzerland (1999–2010) | General (10–39) | 349* | 25344456 | Athletes only | Other | Forensic medicine database | Autopsy record review |
| Bagnall et al[17] | PC | Austrasia (2010–2012) | General (1–3)5 | 490 | 37770000 | No | AHA/ESC | Government records/death certificates; Autopsy records | Autopsy record review |
| Bonny et al[18] | PC | Cameroon (2013) | General (18–39) | – | – | No | Other | Government records/death certificates | Medical record review; interviews |
| Chugh et al[19] | PC | USA (2002–2003) | General (10–34) | – | – | No | AHA/ESC | Hospital/ambulance/medical examiner records; death certificates | Autopsy/medical record review; ICD code |
| Corrado et al[4] | PC | Italy (1979–2004) | General (12–35) | 320 | 36144100 | Athletes only | Other | Medical centre registers, media searches (prospective registry) | Autopsy record review |
| Drehner et al[49] | RC | USA (1956–1996) | Military (17–26) | 23 | 340990 | Yes | Not stated | Military records | Autopsy record review |
| Driscoll and Edwards[44] | RC | USA (1950–1982) | General (1–22) | 7 | 930678 | No | Other | Government records/death certificates | Autopsy/medical record review |
| Eckart et al[43] | RC | USA (1977–2001) | Militrary (17–35) | 64 | – | Yes | Other | Military records | Autopsy/military record review |
| Eckart et al[20] | RC | USA (1998–2008) | Military (<20) | – | – | Yes | Other | Military records | Autopsy record review |
| Einarsson et al[47] | RC | Iceland (1974–2004) | General (12–35) | 42 | – | No | Not stated | Government records/death certificates | Autopsy/medical record review |
| El-Assaad et al[42] | RC | USA (1999–2015) | General (1–34) | 31492 | – | No | Other | Government records/death certificates | ICD code |
| Fragkouli and Vougiouklakis[21] | RC | Greece (1998–2008) | General (1–35) | 28 | – | No | Other | Forensic laboratory records | Autopsy/medical/ police record review |
| Goudevenos et al[41] | PC | Greece (1990–1993) | General (30–39) | – | – | No | Other | Hospital/coroner records; Department of statistics | Autopsy/medical record review; interviews |
| Harmon et al[29] | RC | USA (2003–2013) | Athletes (17–24) | 79 | 4242519 | Yes | AHA/ESC | Media search; NCAA records | Autopsy record review; interviews |
| Harmon et al[30] | RC | USA (2007–2013) | Athletes (14–18) | 69 | 6974640 | Yes | Other | Media search | Autopsy record review |
| Hofer et al[22] | RC | Switzerland (2000–2007) | General (5–39) | 40 | 2340368 | No | Not stated | Government records/death certificates | ICD code |
| Hua et al[46] | PC | China (2005–2006) | General (25–34) | 9 | 115188 | No | Other | Government records/death certificates | Medical record review; interviews |
| Karvouni et al[50] | RC | Greece (1997–1999) | General (1–35) | 33† | 4033333 | No | Other | Forensic laboratory, hospital and ambulance records | Autopsy record review |
| Malhotra et al[23] | PC | UK (1996–2016) | Athletes (15–17) | 8 | 118351 | Yes | Not stated | Voluntary reports/survey; media search | Autopsy record review |
| Margey et al[40] | RC | Ireland (2005–2007) | General (15–34) | 116 | 4065513 | No | AHA/ESC | Government records/death certificates | Autopsy record review |
| Maron et al[31] | RC | USA (2002–2011) | Athletes (17–26) | 64 | 4052369 | No | Not stated | Media search; NCAA records | Autopsy record review |

Continued

**Table 1** Continued

| | Design | Setting | Population: type/age | SCD cases | Follow-up years | Systematic screening | SCD definition | Case ascertainment | |
| --- | --- | --- | --- | --- | --- | --- | --- | --- | --- |
| | | | | | | | | Identification of deaths | Main method to determine SCD |
| Maron et al[32] | RC | USA (2000–2014) | General (14–23) | 27 | 1 308 730 | No | Not stated | Medical examiner records | Autopsy record review |
| Morentin (2001) | RC | Spain (1991–1998) | General (1–35) | 46 | 4 413 088 | No | Other | Forensic laboratory records | Autopsy record review |
| Morentin and Audicana[33] | RC | Spain (2003–2008) | General (35–39) | 42 | – | No | AHA/ESC | Government records/death certificates | Autopsy record review |
| Neuspiel and Kuller[48] | RC | USA (1972–1980) | General (1–21) | 51 | – | No | Other | Government records/death certificates | Autopsy/medical record review |
| Papadakis et al[24] | RC | UK (2002–2005) | General (1–34) | 1677 | 94 256 200 | No | Other | Government records/death certificates | ICD code |
| Perez et al[52] | PC | Spain (1987–1988) | General (25–34) | 1 | – | No | Not stated | Government records/death certificates | Medical record review; interviews |
| Pilmer et al[28] | RC | Canada (2008) | General (2–29) | 56 | – | No | Other | Coroner records | Autopsy/medical/police record review |
| Pilmer et al[27] | RC | Canada (2005–2009) | General (1–19) | 116 | 14 893 860 | No | Other | Coroner records | Autopsy record review |
| Uuskula et al[38] | RC | Estonia (1980–1996) | General (20–39) | 251 | 902 241 | No | Other | Government records/death certificates | Autopsy record review |
| Vaartjes et al[37] | RC | Holland (1996–2006) | General (1–39) | 1500 | 92 374 043 | No | Other | Government records/death certificates | ICD code |
| Winkel et al[26] | RC | Denmark (2000–2009) | General (1–35) | 635 | 23 700 000 | No | AHA/ESC | Government records/death certificates | Autopsy record review |
| Wisten et al[36] | RC | Sweden (1992–1999) | General (15–35) | 181 | 19 514 080 | No | AHA/ESC | Government records/national forensic medicine database | Autopsy/medical record review |
| Wisten et al[25] | RC | Sweden (2000–2010) | General (1–35) | 552 | 42 900 000 | Athletes only | AHA/ESC | Government records/national forensic medicine database | Autopsy record review |
| Wren et al[35] | RC | England (1985–1994) | General (1–20) | 59 | 8 060 000 | No | Other | Coroner/hospital records | Death certificate |
| Zhang et al[34] | RC | China (2015) | General (18–35) | – | – | No | Other | Government records/death certificates | Medical record review; interviews |

*Study inflated number of SCDs to account for low autopsy rate.
†Unable to replicate reported incidence based on study data.
AHA/ESC, American Heart Association/European Society of Cardiology; NCAA, National Collegiate Athletic Association; PC, prospective cohort; RC, retrospective cohort; SCD, sudden cardiac death.

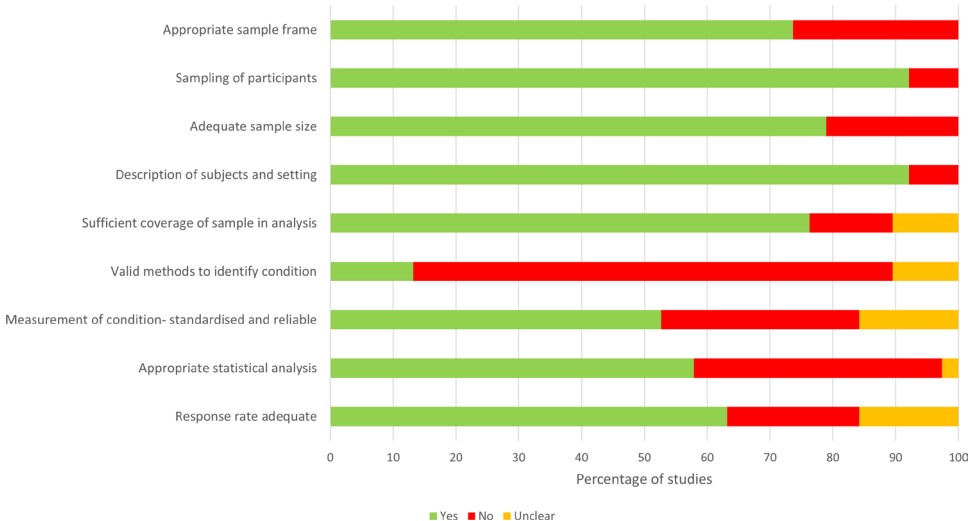

**Figure 2** Risk of bias assessment.

For our subgroup analysis, sex, age, athletic status and race were reported in 13, 11, 9 and 1 studies, respectively. We observed a consistently higher incidence in males (figure 4). In males, median incidence was 2.7 sudden cardiac death per 100 000 person-years (IQR 1.8–4.4, range 1.3–36), compared with a median incidence in females of 0.9 cases per 100 000 person-years (IQR 0.6–1.4, range 0–7.0).

The use of different age cut-offs within studies made a comparison between studies on the association of incidence of sudden cardiac death with age challenging (online supplemental figure S1).

Incidence in military personnel was reported in three studies and appeared to be consistently higher than in other populations (online supplemental figure S2). All three studies were undertaken in American military personnel and reported that individuals received pre-enrolment screening for cardiac conditions. We did not observe a clear difference in incidence between athletic and non-athletic populations. The highest incidence was reported in a prospective study of screened UK soccer players, although the low number of sudden cardiac deaths (eight events) means that the estimate of the incidence is imprecise.

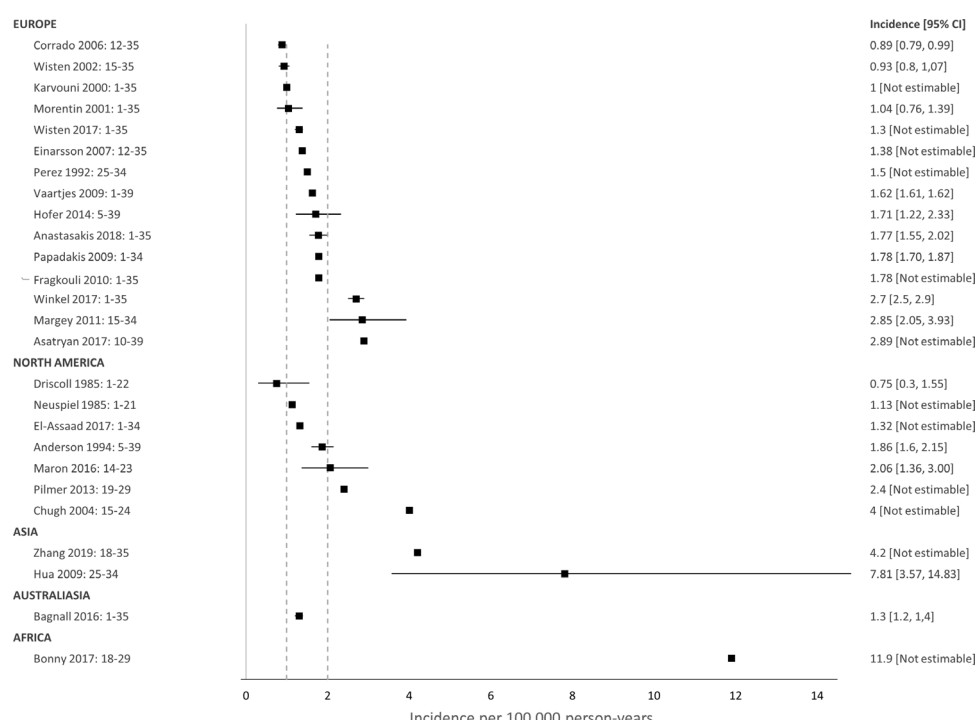

**Figure 3** Incidence of sudden cardiac death in general population. Range following study name indicates population age range.

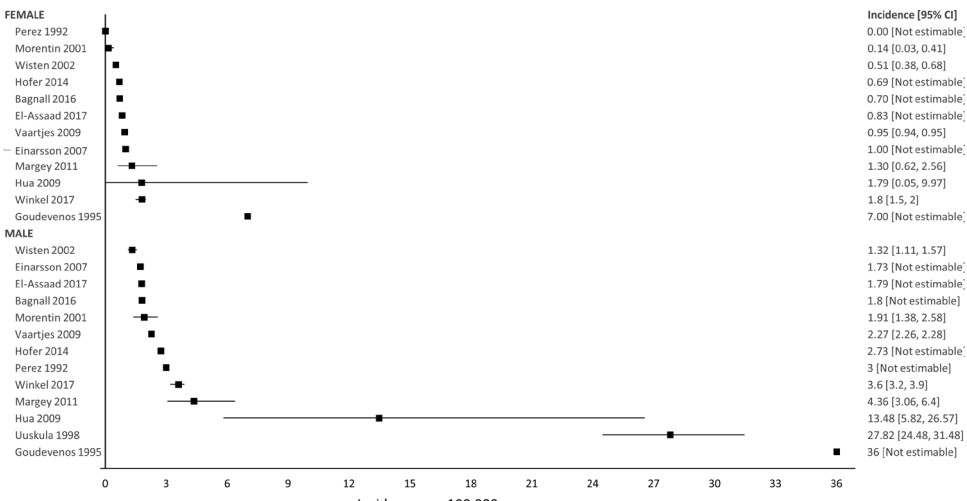

**Figure 4** Incidence of sudden cardiac death in general population-sex subgroups.

Race was reported in a single American study which broke incidence down into four categories (African American, White, Hispanic, other). Incidence was highest in the African-American population (online supplemental figure S3).

## DISCUSSION

In this systematic review, we included 38 studies from five continents that reported the incidence of sudden cardiac death in young individuals. We identified significant variability in the reported incidence across studies. In our main analysis, most studies reported an incidence of between one and two sudden cardiac deaths per 100 000 person-years, although reported incidence ranged from 0.75 to 11.9 cases per 100 000 person-years. This variability may reflect differences in population, case-ascertainment methods, and definition used for sudden cardiac death. In predefined subgroups, we observed that male sex and increasing age seem to be associated with increased incidence.

### Strengths and weaknesses of the study

The key strength of this review was the systematic identification and synthesis of 38 studies from a wide range of settings. We did not restrict inclusion by date and language, with fluent speakers used for translation.

Our review has four key limitations. First, while the geographical area covered by index studies was often large with a long recruitment period, the median number of sudden cardiac deaths identified per study was 64 cases. The method used to identify sudden cardiac death was often sub-optimal, such that misclassification bias may have significantly impacted the reported incidence in some studies.[53] Second, heterogeneity between studies in terms of the population and case ascertainment method used precluded meta-analysis. Thirdly, particularly for sub-groups, studies often described only the absolute incidence, and did not provide sufficient additional data to calculate the 95% CI. As such, we could not describe the

precision of the incidence rate in many studies. Finally, we limited our condition of interest to sudden cardiac death, and did not examine incidence of sudden cardiac arrest. While survival after sudden cardiac arrest is associated with important morbidity, the standard Utstein categorisation of cardiac arrest cause encompasses all medical causes, of which cardiac causes are only a subset.[54 55]

### Comparison with other studies

Our review builds on two previous systematic reviews that described sudden cardiac death incidence across all age groups or in specific groups of young individuals.[10 11] In line with our review findings, these reviews reported issues across index studies in relation to variability in reported incidence, use of different definitions of sudden cardiac death and use of different methods to identify cases of sudden cardiac death.

Variability in reported incidence reflects in part, the challenge of accurately identifying cases of sudden cardiac death. The method used to identify deaths varied across studies, although most used administrative data sources. Nevertheless, the inclusion of some patient groups, such as in-hospital deaths in patients admitted for a minor procedure, may have varied across studies. The definition of sudden cardiac death is based on both the circumstances and cause of death, such that classification requires knowledge of the cause of death and events leading up to the death.[5 6] Only nine studies used the ESC/AHA definition of sudden cardiac death. Other studies either predated this definition of sudden cardiac death or adopted a different definition to fit the method used to identify cases of sudden cardiac death. Death certificate data provide an efficient way to estimate sudden cardiac death incidence. However, their use typically precludes the use of the AHA/ESC definition of sudden cardiac death as the certificate does not include information on the circumstances of the death. As such, studies relying on death certificate data likely overestimate sudden cardiac death incidence.[19]

Determination of the cause of sudden cardiac death usually requires an autopsy, ideally by a specialist pathologist.[56 57] In included studies, reporting of the proportion of patients that received an autopsy and type of autopsy performed was often unclear. This is reflected in our risk of bias assessments. The decision to undertake an autopsy in a specific case may be influenced by factors such as the legal requirement for an autopsy, family acceptance of autopsy and the identification of a likely cause of death without autopsy. A recent European study reported an autopsy rate in young sudden death of less than 50%.[45] Where autopsy is not used, cause of death may be misclassified, although the impact of this misclassification on overall incidence is uncertain. In some cases, cause of death may remain uncertain even after autopsy. In these cases, genetic testing of the deceased may be informative in determining both the cause of death and the need for cardiac screening of the victim's family.[17 57 58] In an Australasian study, genetic testing of 113 cases of unexplained sudden cardiac death cases identified cardiac gene variants that were definitely or probably pathogenic in over a quarter of cases.[17]

Studies included in this review spanned a 66-year period (1950–2016), although most studies report data collected since 2000. Recent observational studies provide evidence of a decreasing incidence of sudden cardiac death.[25 42 59] A Danish study reported a 3% average annual decrease in sudden cardiac death incidence between 2000 and 2009.[59] Potential explanations for this decrease include improved investigation of inherited cardiac disease, public health strategies to deal with obesity and smoking, and improved response to out-of-hospital cardiac arrest.

### Clinical and policy implications

The decision to implement screening programmes in both the UK and USA requires evidence that the benefits of screening outweigh its harms.[9 60] Disease incidence data provides important context for evaluating evidence, particularly in the absence of direct evidence of benefit from randomised controlled trials. In less common conditions, a key concern are the potential harms that may result from false-positive results, and overdiagnosis of disease which may never become symptomatic. Nevertheless, rare disease may still meet criteria for population-based screening conditions, such as maple syrup urine disease in newborns.[61]

The incidence of a condition describes the number of individuals that might benefit from screening. In practice, however, not all young individuals that sustain sudden cardiac death have the potential to benefit from screening.

First, to be effective, screening must reliably identify individuals with disease. In this context, screening must detect a range of structural and electrical cardiac disease. The optimal screening strategy remains uncertain. A wide range of strategies have been described, ranging from those that comprise only a physical examination or medical history to more detailed assessments that include

a 12-lead ECG.[12 62] In programmes where a 12-lead ECG is collected, a number of assessment criteria have been developed to determine the need for follow-up testing.[63–65] In a prospective cohort study of 11 168 screened teenage footballers in which screening incorporated a physical examination, health questionnaire, echocardiogram and 12-lead ECG, six out of eight individuals who subsequently experienced sudden cardiac death did not have disease detected at screening.[23] This highlights a key challenge for policy makers in that, even if they decide to support a screening programme, detailed consideration will need to be given to the screening process and both the financial implications and individual harm that may stem from false-negatives and false-positive screening results.[66 67]

Second, population-level screening is targeted at asymptomatic individuals without a family history of sudden cardiac death. The incidence of sudden cardiac death reported in index studies included both asymptomatic low-risk individuals and individuals at increased risk of sudden cardiac death who, in some countries, may already have access to screening.[6 68–73] A recent study suggested that around one-third of young individuals experienced warning symptoms in the month preceding their sudden cardiac death, indicating the need for clinicians to consider cardiological investigations in symptomatic young individuals.[74] Examples of individuals at increased risk of sudden cardiac death include those with diagnosed cardiac disease, asymptomatic individuals with a family history of sudden cardiac death and symptomatic individuals without diagnosed cardiac disease. As such, policy makers need to be cognisant that incidence reported in population-based studies will overestimate the potential benefit of screening programmes which are targeted at low-risk asymptomatic individuals.

Finally, effective screening relies on the individuals willingness to be screened. To date, most studies report screening which, while technically voluntary, was mandated for participation in an activity, such as sport. Studies of athletes report young people are supportive of screening.[75 76] In practice, however, uptake of school-based cardiac screening programmes varies from 56% to 79%, with substantial variation by year and school.[77 78]

In conclusion, studies of incidence of sudden cardiac death in young individuals have produced varying estimates of the incidence, although most studies report the incidence to be between one and two cases per 100 000 person-years. In these studies of young individuals, incidence was highest in males and among older individuals.

**Contributors** KC accepts full responsibility for the work and conduct of the study, had access to the data and controlled the decision to publish. KC designed the study, acted as reviewer, undertook the analysis and drafted the manuscript. OP and RF acted as reviewers. PR designed database searches. KP, WB, AC, GP, JY and ST-P contributed to study design. All authors contributed to the write up and approved the final version.

**Funding** This work was supported by Public Health England (Award/Grant number is not applicable). Public Health England originally commissioned a rapid review through the UK National Screening Committee, on which this systematic

review was based. Otherwise, Public Health England had no role in study design; collection, analysis and interpretation of data; in the writing of the report and in the decision to submit the article for publication. Keith Couper had full access to all of the data (including statistical reports and tables) in the study and can take responsibility for the integrity of the data and the accuracy of the data analysis. This work was supported by the National Institute for Health Research (NIHR) Applied Research Collaboration (ARC) West Midlands (Award/Grant number is not applicable). Professor Perkins is supported as an NIHR senior investigator. Professor Taylor-Phillips is supported by an NIHR Career Development Fellowship (CDF-2016-09-018). The opinions are those of the authors and not the NIHR, the UK National Screening Committee, the NHS or the Department of Health and Social Care.

**Competing interests** All authors have completed the ICMJE uniform disclosure at www.icmje.org/coi_disclosure.pdf and declare: KC, KP, AC, PR, GP, JY, ST-P received financial support from Public Health England for the rapid review on which this review was based; no financial relationships with any organisations that might have an interest in the submitted work in the previous 3 years; KC, JY and GP have volunteer roles with the International Liaison Committee on Resuscitation, European Resuscitation Council and Resuscitation Council UK.

**Patient and public involvement** Patients and/or the public were not involved in the design, or conduct, or reporting, or dissemination plans of this research.

**Patient consent for publication** Not required.

**Provenance and peer review** Not commissioned; externally peer reviewed.

**Data availability statement** All data relevant to the study are included in the article or uploaded as supplement information. All data relevant to the study are included in the article or uploaded as supplement information.

**Open access** This is an open access article distributed in accordance with the Creative Commons Attribution 4.0 Unported (CC BY 4.0) license, which permits others to copy, redistribute, remix, transform and build upon this work for any purpose, provided the original work is properly cited, a link to the licence is given, and indication of whether changes were made. See: https://creativecommons.org/licenses/by/4.0/.

**ORCID iDs**
Keith Couper http://orcid.org/0000-0003-2123-2022
Joyce Yeung http://orcid.org/0000-0003-2950-4758
Sian Taylor-Phillips http://orcid.org/0000-0002-1841-4346

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
