## [Reviewer comments · BMJ Open]

ARTICLE DETAILS

TITLE (PROVISIONAL)	Incidence of sudden cardiac death in the young: a systematic review
AUTHORS	Couper, Keith; Putt, Oliver; Field, Richard; Poole, Kurtis; Bradlow, William; Clarke, Aileen; Perkins, Gavin; Royle, Pamela; Yeung, Joyce; Taylor-Phillips, Sian

VERSION 1 – REVIEW

REVIEWER	Thomas Hadberg Lyng Copenhagen University Hospital, Rigshospitalet, The Heart Centre, Denmark
REVIEW RETURNED	06-Jul-2020

GENERAL COMMENTS	The authors have completed a comprehensive review describing the incidence of sudden cardiac death in the young general population. The manuscript is well-written. The methods and results are adequately described, and the authors provide an excellent overview of the existing information on SCD in the young. The information provided in the review is important and this reviewer completely agrees that information on SCD epidemiology is key in developing novel and improving existing preventive strategies aimed at reducing the burden of SCD in the general population. The objective of the paper was “To summarise studies describing incidence of sudden cardiac death in a general population of young individuals to inform screening policy.” While this reviewer fully agrees that the authors provide an excellent and comprehensive overview of the incidence of SCD in the young, the main objections is that the review provides limited (new) information to guide/inform screening policy. The discussions sections should be considerably expanded to address how the information collected in this review and the conclusions reached inform screening policy. Otherwise that part should be omitted from the objectives section. Other comments: 1) There is growing evidence supporting that many SCD cases (both young and among cases of all ages) exhibit symptoms prior to death and that many of these contact the healthcare system on basis of these symptoms. This is obviously interesting as these cases represents missed opportunities for prevention and cardiac symptoms could in the future be the basis of tailored screening of persons in moderate-to-high risk of SCD as opposed to general screening of the (asymptomatic) general population. The authors are asked to comment/expand the relevant discussions section. 2) Please provide information as to why an age cut-off of 12 years was chosen. The authors state that they include studies where the reported incidence combined individuals in their target range (12-39 years) with younger individuals > 1 year. When cases aged 1-12
---

	years are included in some instances, why not systematically include all cases aged 1-39 years? The results could be subcategorized into age groups in additional analyses. 3) The authors state that the risk of bias in many studies was high. The authors state that a Joanna Briggs Institute critical appraisal checklist for prevalence studies was used. Please provide further information as to how this was used and how risk of bias was evaluated/measured. E.g. a more detailed/specific discussion of why the studies reporting high SCD incidence were in high risk of bias would add to the paper. 4) The authors state that differences in methodological and definitional approaches likely explain the rather large variations in reported incidences. It is, however, not stated how these differences specifically explain the differences in reported incidence. A more comprehensive discussion of this subject would add to the paper. 5) There has been some discussion as to whether in-hospital sudden death (e.g. in patients admitted for minor conditions or for examinations of symptoms such as seizures or syncopes) should be included when reporting incidence of SCD. The authors are asked to comment/address this subject in the review. 6) The authors state that an autopsy is usually necessary to determine cause of death. In many studies of SCD incidence in the young, the autopsy rate is far from 100%. Could the authors please comment/discuss further how differences in autopsy rates affect reported SCD incidences? 7) Out of curiosity: The authors had no restrictions on language in their literature search. Did they hire external fluent speakers?
--	--

REVIEWER	Elizabeth Paratz Baker Heart & Diabetes Institute
REVIEW RETURNED	17-Jul-2020

GENERAL COMMENTS	This is a well-written and thorough review. Some minor comments: # Why was the very specific age range of 12-39 years chosen? This is never clearly defined. There is good justification of why not <1 year old but why not 1-11 years old? Particularly as multiple studies enrolling patients aged 1-11 years old have then been included 'for reasons of pragmatism', would it not be better to just have a range of 1-39 years? # One of the key limitations not commented on is that using forensic results to estimate incidence will inevitably lead to under-estimation of the incidence of SCD. In many countries, autopsies are progressively less common (even though recommended in young sudden death). In the Bonny paper for example, no autopsies at all were conducted in the study period. True estimation of the incidence of SCD requires a multi-source surveillance approach, with limitations apparent to all forms of single-site sampling. # When discussing 'screening programmes', it is never clearly defined what is implied by screening - for example, ECG and echo or ECG alone? Important to comment that there is a range of aetiologies causing sudden cardiac death, with most common cause in young now being 'unascertained' even after comprehensive autopsy assessment (Bagnall reference). These cases wouldn't be picked up by any pre-mortem screening program but confer the majority of SCD burden in the young. # When discussing the public health risks/benefits of a screening
--

	program, a brief mention / estimate of the economic impact of screening programmes and lack of utility would also be valuable. # There is significant variation in the referencing: for example some titles are extensively capitalized, some not. Suggest standard approach according to journal recommendations
--	---

VERSION 1 – AUTHOR RESPONSE

Reviewer: 1

Reviewer Name: Thomas Hadberg Lyngø

Institution and Country: Copenhagen University Hospital, Rigshospitalet, The Heart Centre, Denmark

Competing interests: None declared

Please leave your comments for the authors below

The authors have completed a comprehensive review describing the incidence of sudden cardiac death in the young general population. The manuscript is well-written. The methods and results are adequately described, and the authors provide an excellent overview of the existing information on SCD in the young. The information provided in the review is important and this reviewer completely agrees that information on SCD epidemiology is key in developing novel and improving existing preventive strategies aimed at reducing the burden of SCD in the general population.

>Thank you for this summary of our paper.

The objective of the paper was “To summarise studies describing incidence of sudden cardiac death in a general population of young individuals to inform screening policy.” While this reviewer fully agrees that the authors provide an excellent and comprehensive overview of the incidence of SCD in the young, the main objection is that the review provides limited (new) information to guide/inform screening policy. The discussions sections should be considerably expanded to address how the information collected in this review and the conclusions reached inform screening policy. Otherwise that part should be omitted from the objectives section.

>We are grateful to the reviewer for highlighting that our review provides a detailed and important overview of SCD incidence in the young. The reviewer is correct that heterogeneity across studies of SCD incidence precludes pooling of data- whilst some might view this as a limitation, we feel that this is an extremely important finding for policy makers. If we are unable to estimate the potential benefit of a screening programme, it becomes extremely challenging for a policy maker to balance potential benefit against harm. In our discussion, we already highlight the incidence of sudden cardiac death reflects the maximum benefit that may be derived from a screening programme. We have expanded this section considerably to provide further context for policy makers.

Other comments:

1) There is growing evidence supporting that many SCD cases (both young and among cases of all ages) exhibit symptoms prior to death and that many of these contact the healthcare system on basis of these symptoms. This is obviously interesting as these cases represent missed opportunities for prevention and cardiac symptoms could in the future be the basis of tailored screening of persons in moderate-to-high risk of SCD as opposed to general screening of the (asymptomatic) general population. The authors are asked to comment/expand the relevant discussions section.

>Thank you- we have added a comment to the discussion section.

2) Please provide information as to why an age cut-off of 12 years was chosen. The authors state that they include studies where the reported incidence combined individuals in their target range (12-39 years) with younger individuals 1 year. When cases aged 1-12 years are included in some instances, why not systematically include all cases aged 1-39 years? The results could be subcategorized into age groups in additional analyses.

>Thank you- the age cut-off was prospectively defined. In the UK, the age group of 12-39 was identified as a target age range for consideration of a screening programme by the UK National Screening Committee. In considering age ranges, there is a challenging trade-off between including as much data as possible and ensuring that the data included is relevant to the research question and of value to policy makers. We have explained our strategy in the methods. In our results, we do report the incidence by age, but comparison between studies is challenging as studies use different age cut-offs.

3) The authors state that the risk of bias in many studies was high. The authors state that a Joanna Briggs Institute critical appraisal checklist for prevalence studies was used. Please provide further information as to how this was used and how risk of bias was evaluated/measured. E.g. a more detailed/specific discussion of why the studies reporting high SCD incidence were in high risk of bias would add to the paper.

>Thank you- we have more detail in the results section. The methods section details how the risk of bias was assessed. In the supplementary information, we provide a detailed assessment of all included studies.

4) The authors state that differences in methodological and definitional approaches likely explain the rather large variations in reported incidences. It is, however, not stated how these differences specifically explain the differences in reported incidence. A more comprehensive discussion of this subject would add to the paper.

>Thank you for this comment. We have expanded this point in the discussion.

5) There has been some discussion as to whether in-hospital sudden death (e.g. in patients admitted for minor conditions or for examinations of symptoms such as seizures or syncopes) should be included when reporting incidence of SCD. The authors are asked to comment/address this subject in the review.

>Thank you- we have briefly covered this point in the discussion.

6) The authors state that an autopsy is usually necessary to determine cause of death. In many studies of SCD incidence in the young, the autopsy rate is far from 100%. Could the authors please comment/discuss further how differences in autopsy rates affect reported SCD incidences?

>Thank you- this is an important point, but challenging to determine. Most studies did not detail the proportion of individuals that received an autopsy. We have included some additional detail in the discussion.

7) Out of curiosity: The authors had no restrictions on language in their literature search. Did they hire external fluent speakers?

>Yes we did- we have clarified this in the methods. This point was already noted in the discussion.

Reviewer: 2

Reviewer Name: Elizabeth Paratz

Institution and Country: Baker Heart & Diabetes Institute

Competing interests: none

Please leave your comments for the authors below

This is a well-written and thorough review.

>Thank you.

Some minor comments:

Why was the very specific age range of 12-39 years chosen? This is never clearly defined. There is good justification of why not <1 year old but why not 1-11 years old? Particularly as multiple studies enrolling patients aged 1-11 years old have then been included 'for reasons of pragmatism', would it not be better to just have a range of 1-39 years?

>Thank you for flagging this important point. We have provided clarification both in the text and in the response to the previous reviewer.

One of the key limitations not commented on is that using forensic results to estimate incidence will inevitably lead to under-estimation of the incidence of SCD. In many countries, autopsies are progressively less common (even though recommended in young sudden death). In the Bonny paper for example, no autopsies at all were conducted in the study period. True estimation of the incidence of SCD requires a multi-source surveillance approach, with limitations apparent to all forms of single-site sampling.

>Thank you- we agree that autopsy (ideally supplemented by genetic testing) is the most robust way to determine cardiac cause of death. The Bonny paper does note in the limitation section that use of autopsy to determine cause of death was not systematic, so it appears that some individuals received an autopsy. We have included some additional reflections on autopsy in the discussion.

When discussing 'screening programmes', it is never clearly defined what is implied by screening - for example, ECG and echo or ECG alone? Important to comment that there is a range of aetiologies causing sudden cardiac death, with most common cause in young now being 'unascertained' even after comprehensive autopsy assessment (Bagnall reference). These cases wouldn't be picked up by any pre-mortem screening program but confer the majority of SCD burden in the young.

>Thank you- we note in the discussion that screening must identify a range of electrical and structural disease. We have added some detail about screening programmes.

When discussing the public health risks/benefits of a screening program, a brief mention / estimate

of the economic impact of screening programmes and lack of utility would also be valuable.

>We have more detail in the results section. The methods section details how the risk of bias was assessed. In the supplementary information, we provide a detailed assessment of all included studies. We have added a comment about the economic impact of screening.

There is significant variation in the referencing: for example some titles are extensively capitalized, some not. Suggest standard approach according to journal recommendations

>Thank you for noting this- our capitalisation reflects the capitalisation in the original publication. Unfortunately, from the journal author instructions, it is a little unclear what the journal style is. If accepted for publication, we are happy to work with the copywriting team to ensure that the paper complies with the journal style.

VERSION 2 – REVIEW

REVIEWER	Thomas Hadberg Lyng Copenhagen University Hospital, Rigshospitalet. The Heart Centre. Denmark
REVIEW RETURNED	25-Aug-2020

GENERAL COMMENTS	I have no further comments.
-----------------------------

REVIEWER	Elizabeth Paratz Baker Heart & Diabetes Institute, Australia
REVIEW RETURNED	23-Aug-2020

GENERAL COMMENTS	The authors have addressed the reviewers' key questions. One small amendment I would recommend prior to acceptance is the statement that 'in Australia the rate of autopsy in young sudden cardiac death is 100% (citing reference 17). This was a study that recruited subjects from forensic institutions, hence 100% amongst the subset of young SCD patients undergoing autopsy. The rate of autopsy in young SCD will always be <100% due to family objections, reasonable cause of death being obtained without autopsy, missed recognition of case etc.
--

VERSION 2 – AUTHOR RESPONSE

Reviewer: 2

Reviewer Name: Elizabeth Paratz

Institution and Country: Baker Heart & Diabetes Institute, Australia

Competing interests: None declared

Please leave your comments for the authors below

>The authors have addressed the reviewers' key questions. One small amendment I would recommend prior to acceptance is the statement that 'in Australia the rate of autopsy in young sudden cardiac death is 100% (citing reference 17). This was a study that recruited subjects from forensic institutions, hence 100% amongst the subset of young SCD patients undergoing autopsy. The rate of

autopsy in young SCD will always be <100% due to family objections, reasonable cause of death being obtained without autopsy, missed recognition of case etc.

Thank you- we have removed the comment about the autopsy rate of 100% in case it might be misleading. We have added some text regarding factors that might influence autopsy rate.